User-centric AI: evaluating the usability of generative AI applications through user reviews on app stores

Alabduljabbar Reham ralabduljabbar@ksu.edu.sa
Information Technology Department, College of Computer and Information Sciences, King Saud University , Riyadh , Saudi Arabia
Balas Valentina Emilia
Electronic publication date: 2024 Oct 25
Publication date: 2024
Volume: 10
Electronic Location ID: e2421
Received 2024 May 6; Accepted 2024 Sep 25
Copyright: ©2024 Alabduljabbar
Copyright year: 2024
Copyright holder: Alabduljabbar
License: This is an open access article distributed under the terms of the Creative Commons Attribution License, which permits unrestricted use, distribution, reproduction and adaptation in any medium and for any purpose provided that it is properly attributed. For attribution, the original author(s), title, publication source (PeerJ Computer Science) and either DOI or URL of the article must be cited.
License URL: https://creativecommons.org/licenses/by/4.0/

Keywords: User-centred design, Usability evaluations, LLMs, Generative AI, User experience, User review, ChatGPT, Microsoft Copilot, HCI, Human–AI interaction

Funding: Researchers Supporting Project RSPD2024R905 This research project was supported by Researchers Supporting Project number (RSPD2024R905), King Saud University, Riyadh, Saudi Arabia. The funders had no role in study design, data collection and analysis, decision to publish, or preparation of the manuscript.

==============================
This article presents a usability evaluation and comparison of generative AI applications through the analysis of user reviews from popular digital marketplaces, specifically Apple’s App Store and Google Play. The study aims to bridge the research gap in real-world usability assessments of generative AI tools. A total of 11,549 reviews were extracted and analyzed from January to March 2024 for five generative AI apps: ChatGPT, Bing AI, Microsoft Copilot, Gemini AI, and Da Vinci AI. The dataset has been made publicly available, allowing for further analysis by other researchers. The evaluation follows ISO 9241 usability standards, focusing on effectiveness, efficiency, and user satisfaction. This study is believed to be the first usability evaluation for generative AI applications using user reviews across digital marketplaces. The results show that ChatGPT achieved the highest compound usability scores among Android and iOS users, with scores of 0.504 and 0.462, respectively. Conversely, Gemini AI scored the lowest among Android apps at 0.016, and Da Vinci AI had the lowest among iOS apps at 0.275. Satisfaction scores were critical in usability assessments, with ChatGPT obtaining the highest rates of 0.590 for Android and 0.565 for iOS, while Gemini AI had the lowest satisfaction rate at −0.138 for Android users. The findings revealed usability issues related to ease of use, functionality, and reliability in generative AI tools, providing valuable insights from user opinions and feedback. Based on the analysis, actionable recommendations were proposed to enhance the usability of generative AI tools, aiming to address identified usability issues and improve the overall user experience. This study contributes to a deeper understanding of user experiences and offers valuable guidance for enhancing the usability of generative AI applications.

Introduction

The field of generative AI has experienced significant global growth, with a diverse range of applications expanding rapidly. Research publications and comparative analyses have been instrumental in enhancing our understanding of the capabilities of generative AI tools (Iorliam & Ingio, 2024). Just as user feedback is crucial for mobile app enhancements, evaluations and reviews of generative AI tools are critical for their refinement (Rizk, Ebada & Nasr, 2015). They provide insight into user satisfaction and can drive the development of more user-friendly and efficient AI systems.

Generative AI, encompassing tools like ChatGPT, has revolutionized the way we interact with machine-generated content, offering capabilities from text generation to creating complex data models. User interactions with these tools, through reviews and feedback, often yield valuable insights that can lead to improvements in functionality and usability (Dey, 2024).

Despite the growing interest and impact of generative AI (Precedence Research, 2024), there exists a noticeable gap in conducting comprehensive analyses of the usability of these tools. The absence of thorough comparative analyses impedes the selection of the most suitable generative AI tool for specific applications. Usability, which includes aspects like ease of use and efficiency, is critical for determining the applicability and effectiveness of generative AI tools in real-world scenarios.

Designing user experiences for generative AI applications poses distinct challenges. With the integration of generative AI technologies into common applications on the rise, there is an urgent need for guidance on how to design user experiences that promote effective and secure utilization (Weisz et al., 2024). In the study, six principles were developed for designing generative AI applications, addressing unique characteristics of generative AI UX and known issues in AI application design. The principles aim to provide actionable design recommendations for informing the design of generative AI applications effectively.

Studies such as Bubaš, Čižmešija & Kovačić (2024) have emphasized the significance of usability and user experience characteristics in predicting users’ intentions to use AI applications like Bing Chat or ChatGPT. The research identified usability factors like perceived usefulness, as well as user experience elements such as trust and design appeal, as crucial in shaping users’ future adoption intentions. Similarly, Mulia, Piri & Tho (2023) conducted a usability analysis of text generation by ChatGPT OpenAI, highlighting the importance of employing systematic usability assessment methods. Most of the existing studies, although limited in number, have focused on usability analysis by examining datasets gathered from social media or employing usability testing methods such as the System Usability Scale (SUS) and the User Experience Questionnaire (UEQ) (Alhejji et al., 2024). However, while these studies provide valuable insights into usability and user experience aspects, there remains a gap in understanding how users perceive generative AI applications in real-world settings. Little attention was given to the huge amounts of reviews for generative AI applications on Google Play or the App Store. Therefore, evaluating generative AI apps from users’ perspectives in commercial mobile applications, as proposed in the current research Bubaš, Čižmešija & Kovačić (2024) and Mulia, Piri & Tho (2023), can offer a unique and practical understanding of user preferences and concerns, bridging the existing gap in academic literature by focusing on users’ opinions from platforms like Google Play or the App Store. Additionally, the reviews directly reported by app users are considered more reliable than those from other sources, such as social media, which can be a source of rumors or disinformation (Alghareeb, Albesher & Asif, 2023; Hadwan et al., 2022).

To address this gap, this article aims to assess and compare the usability of various generative AI tools guided by ISO 9241’s usability standards, focusing on effectiveness, efficiency, and satisfaction. By analyzing user reviews from five generative AI apps, this research seeks to identify crucial usability issues and propose actionable solutions for developers to enhance the usability of these tools. Specifically, this article focuses on analyzing user reviews from five generative AI apps. The study utilized sentiment analysis to evaluate English reviews, allowing for the processing of large datasets without the need for participant identification or specific analytical tools like questionnaires, which are typically required by other methods such as SUS and UEQ. Data collection involved the use of Python scripts and web scraping tools to gather 11,549 reviews from Android and iOS platforms between January and March 2024. The reviews were publicly available.

Ultimately, the research aims to offer recommendations for enhancing the usability of generative AI tools, thereby contributing to informed decision-making for developers, researchers, and businesses relying on generative AI solutions.

The rest of the article is organized as follows. ‘Background’ provides the necessary background information on the topic. ‘Related Works’ xplores the relevant literature related to generative AI usability. ‘Materials and Methods’ explains the research methodology adopted in the study. ‘Results and Analysis’ interprets the collected data, discusses the findings, and addresses limitations and future work. This section also consolidates the main usability challenges observed in generative AI tools and offers recommendations for improvement. ‘Conclusions’ presents the concluding remarks of the article.

Background

Generative AI tools

Generative AI, a field at the forefront of artificial intelligence research, focuses on creating new content rather than analyzing existing data. This innovative technology leverages generative models, such as generative adversarial networks (GANs) and large language models (LLMs), to generate diverse outputs like images, text, and music. GANs, a prominent technique in generative AI is the generative adversarial network (GAN), which consists of two neural networks, a generator, and a discriminator, that work in tandem to produce realistic outputs. LLMs, another essential tool in generative AI, excel in generating human-like text and have applications across various domains. However, challenges like bias and misinformation persist, as LLMs may inadvertently produce content that is factually inaccurate or perpetuates harmful stereotypes (Feuerriegel et al., 2024).

The field of generative AI is continuously evolving, with implications across various sectors. In healthcare, generative AI is being explored for applications like creating conversational agents for chronic conditions to provide support and information to patients (Schachner, Keller & Wangenheim, 2020). In marketing, the use of generative AI raises both opportunities and pitfalls, as organizations leverage AI solutions for higher-order learning but also face challenges in adoption and implementation (De Bruyn et al., 2020). As generative AI progresses, considerations around governance and ethical implications become crucial. Discussions on global cooperation on AI governance highlight the need for frameworks that address the why, what, and how of governing AI technologies (Ala-Pietilä & Smuha, 2021). Moreover, the potential risks associated with AI, including its impact on existential factors, are subjects of research and debate, emphasizing the importance of understanding and mitigating these risks (Bucknall & Dori-Hacohen, 2022).

Some of the leading tools include GPT-4 (ChatGPT) by OpenAI (https://openai.com/gpt-4), Microsoft Copilot (https://www.microsoft.com/en/copilot), Gemini (formerly Bard) by Google (https://gemini.google.com), GitHub Copilot (https://github.com/features/copilot), Claude by Anthropic (https://claude.ai/chats), Cohere Generate (Command) by Cohere (https://cohere.com), Synthesia (http://synthesia-8269203.hs-sites.com/hs-web-interactive-8269203-153870024198), DALL-E 3 (https://openai.com/dall-e-3) by OpenAI, Midjourney (https://www.midjourney.com/home), and Jasper AI (https://www.jasper.ai). These tools cater to various needs such as content generation, coding quality assurance, ethical content generation, video creation, image generation, digital marketing content generation, and more.

GPT-4 (ChatGPT) is highlighted as a popular tool known for its large language model and content generation capabilities. Microsoft Copilot is recognized for its AI copilot features for general business use, while Gemini (formerly Bard) stands out for real-time online resources and connectivity. GitHub Copilot by Microsoft aims at coding quality assurance with subscription-based pricing. Claude is praised for its focus on ethical and secure business content generation and Cohere Generate (Command) is commended for its straightforward API integration. Synthesia is noted for AI-powered video creation, and DALL-E 3 excels in accessible image generation. Midjourney is recognized for advanced AI image editing and generation, and Jasper AI is highlighted for digital marketing content generation.

User reviews and sentiment analysis

User reviews are a crucial component of today’s digital landscape, providing valuable insights into products, services, and experiences. These reviews, commonly found on platforms such as app stores, e-commerce websites, and social media, offer customers a space to voice their opinions, share feedback, and rate their interactions with businesses. They often include ratings, written comments, and other qualitative or quantitative assessments, covering various topics such as user experience, technical issues, and feature suggestions (Pagano & Maalej, 2013). Researchers have utilized user reviews to gain deeper insights into app performance and user experiences. For instance, Di Sorbo et al. (2021) emphasize the significance of user reviews in providing valuable comments, bug reports, and feature requests that guide developers in meeting user needs

Accessing user reviews is relatively straightforward, as they are publicly available on numerous online platforms, allowing potential users to evaluate them before making purchasing decisions or using apps.

User reviews significantly influence consumer behavior and decision-making processes (Son et al., 2020). They offer prospective users insights into product performance, usability, quality, and reliability, playing a crucial role in shaping purchasing decisions (Jin, DiPietro & Watanabe, 2022). Positive reviews can foster trust, boost user adoption, and drive app downloads, while negative reviews can deter potential users and highlight areas for improvement.Analyzing user reviews can lead to enhanced app design, functionality, and user satisfaction.

Sentiment analysis, also known as opinion mining, is a sub-field of natural language processing (NLP) that aims to determine the sentiment behind a series of words used to express opinion or emotion, particularly in online text. Sentiment analysis of user reviews has proven invaluable in understanding user sentiments and preferences across various industries (Pandey & Kumar, 2021), enabling the determination of the general tone of consumer reviews (Al-Natour & Turetken, 2020). This analytical approach leverages AI, computational linguistics, and text analysis to identify subjective information, ranging from detecting polarity (positive/negative) to recognizing emotions (happy, sad, angry) and intentions (interested/not interested) (Hossain et al., 2023).

Various methods can be employed for sentiment analysis, including lexicon-based approaches, machine learning algorithms, and deep learning models (Nasukawa & Yi, 2003). Lexicon-based methods utilize predefined sentiment lexicons to assign sentiment scores to words and phrases, while machine learning algorithms learn patterns from labeled data to classify sentiment. Deep learning models, such as recurrent neural networks (RNNs) and convolutional neural networks (CNNs), have shown promise in sentiment analysis tasks due to their ability to capture intricate patterns in text data.

Analyzing user reviews enables researchers to quantitatively summarize key themes and qualitatively extract rich insights into users’ viewpoints (Karim et al., 2020), which are essential for app developers to enhance app functionality and user experience. While user reviews provide a wealth of information, challenges such as opinion spam and deceptive reviews can potentially mislead the analysis (Tangari et al., 2021). Nonetheless, publicly available app reviews and ratings offer a wealth of information regarding user perspectives and usability issues (Aydin & Silahtaroglu, 2021; Balaskas et al., 2022).

Usability

The International Organization for Standardization (ISO) defines usability as “the extent to which a product can be used by specified users to achieve specified goals with effectiveness, efficiency, and satisfaction in a specified context of use” (ISO 9241-11).

The standard outlines three key factors that influence usability: effectiveness, efficiency, and satisfaction (ISO, 2024).

ISO 9241-11 defines these factors as follows:

• Effectiveness: the accuracy and completeness with which users achieve specified goals.

• Efficiency: the resources used in relation to the results achieved.

• Satisfaction: the extent to which the user’s physical, cognitive, and emotional responses that result from the use of a system, product, or service meet the user’s needs and expectations.

In the context of generative AI tools, usability is paramount in ensuring that users can effectively interact with and derive value from these tools. The usability of generative AI tools is crucial in determining their adoption and success among users. Factors such as ease of use, intuitive design, and task efficiency play a significant role in enhancing the usability of these tools. By focusing on usability, developers can create tools that are user-friendly, efficient, and align with user expectations, ultimately leading to improved user experiences and satisfaction. The failure of apps and software often stems from the inability of a system to meet users’ goals and measure their satisfaction. As a result, assessing usability has become an integral part of app and software development (Park & Zahabi, 2021). Usability testing methods, such as System Usability Scale (SUS), the User Experience Questionnaire (UEQ), heuristic evaluation, and cognitive walkthroughs, are commonly employed to assess the usability (Alhejji et al., 2024). The SUS is currently one of the most commonly used tools for usability testing. It was designed to provide a quick and straightforward way to evaluate usability (Brooke, 1995).

These methods help identify usability issues, gather user feedback, and make informed design decisions to enhance the overall usability of the tools. By prioritizing usability in the design and development of generative AI tools, businesses can ensure that these tools are accessible, efficient, and user-centric, leading to enhanced user experiences and increased user satisfaction. Current research explores the usability of early generative AI tools, seeking to uncover user behaviors and identify usability issues that could impede the adoption of such technology (Mugunthan, 2024).

Related Works

Generative AI tools have been the subject of extensive usability research in recent years. Studies have delved into various aspects of these tools, aiming to evaluate their effectiveness and user-friendliness.

Studies on conversational AI like ChatGPT have focused on enhancing usability for effective user interactions. Pardos & Bhandari (2023) emphasized the usability improvements in ChatGPT, attributing them to an intuitive interface and the evolution of GPT models. This evolution involved human raters to enhance the model’s text generation alignment with desirable responses, showcasing advancements in usability. Studies by Ren et al. (2022) and Ren et al. (2024) provided insights into the experimentation and usability evaluation of chatbots, aiming to enhance real-time collaboration tools and user-perceived usability. These studies contribute to the understanding of how chatbots can be optimized for effective user interactions. In addition, the study in Bubaš, Čižmešija & Kovačić (2024) introduces a new set of assessment scales related to the usability and user experiences of conversational artificial intelligence (CAI) tools among university students, focusing on the evaluation of the Bing Chat tool in educational settings. The scales cover various aspects like perceived usefulness, general usability, and more. The research finds that students’ intention to use these CAI tools is significantly influenced by how useful they find the tool, the trust they have in it, and its design appeal. Similarly, the work in Mulia, Piri & Tho (2023) focuses on evaluating the quality and usability of ChatGPT to ensure a satisfactory user experience. Researchers utilized the System Usability Scale (SUS) to quantitatively assess ChatGPT, conducting an online survey to collect data. Upon validation, the analysis yielded a SUS score of 67.44 for ChatGPT, categorizing it in the ’marginal high’ region of Class D. This indicates a generally positive user response to its effectiveness and efficiency, as well as ease of use and understanding. It also highlights a limitation regarding the completeness and currency of information, as ChatGPT’s knowledge base is confined to data up to the year 2021. The article suggests that further studies should explore additional functionalities of ChatGPT to ascertain the system’s comprehensive stability and reliability. Another study Skjuve, Følstad & Brandtzaeg (2023) also explores the user experience of ChatGPT, highlighting its significant impact in the realm of conversational AI. Through a survey of 194 ChatGPT users, the research delves into both positive and negative feedback. Analysis via a pragmatic-hedonic framework reveals that user satisfaction with ChatGPT stems from its practicality, like providing relevant information and facilitating academic or professional tasks. Additionally, the hedonic aspects, such as enjoyment and innovative engagement, also play a crucial role in shaping user experiences. Furthermore, Lin (2023) highlighted how conversational AI, like chatbots, can enhance system usability by providing intuitive interfaces for user input, emphasizing the importance of user-friendly designs in these tools.

Generative AI’s role in specialized domains has been explored by studies like (Rodriguez et al., 2024), where ChatGPT contributed to a digital health intervention, demonstrating its utility in medical technology development. The study involved 11 evaluators with expertise in various fields evaluating ChatGPT-generated outputs in terms of understandability, usability, novelty, relevance, completeness, and efficiency. The results indicated that ChatGPT can expedite the development of high-quality products and enhance communication between technical and non-technical team members in creating computational solutions for medical technologies. The findings suggest that ChatGPT can effectively support researchers throughout the software development life cycle, from conceptualization to code generation, demonstrating its potential as a facilitator in product development processes. This is complemented by research Bandi, Adapa & Kuchi (2023) investigating the key aspects of generative AI, including requirements, models, input–output formats, and evaluation metrics. It categorizes the requirements for generative AI systems into hardware, software, and user experience. The study presents a taxonomy of generative AI models based on architectural features, covering VAEs, GANs, diffusion models, transformers, language models, normalizing flow models, and hybrid models. Additionally, it offers a detailed classification of input–output formats and discusses evaluation metrics used in generative AI. The findings help in implementing and assessing generative AI models effectively, highlighting the importance of understanding system requirements, selecting appropriate models, utilizing diverse formats, and employing standardized evaluation methods for optimal performance and advancements in the field.

The usability of ambient and mobile interfaces has been assessed through various studies such as the work by Romeiro & Araújo (2024), which defined guideline-based metrics for evaluating Ambient Assisted Living (AAL) ecosystem usability, providing tools for developers to assess usability bottlenecks accurately. Their work contributes to enhancing the usability assessment of mobile interfaces through intelligent optimization approaches. In addition, Sally (2023) conducted a text mining analysis on Sri Lankan mobile banking apps to understand consumer dissatisfaction. The study highlighted the importance of sentiment analysis in evaluating user opinions and improving services based on user feedback. Alshamari & Alsalem (2023) conducted a critical review of current issues and trends in usable AI, emphasizing the importance of preliminary usability evaluations for cognitive systems powered by AI. This study underscores the need for standardized usability measures and further research in AI usability. Tao et al. (2023) explored the maintenance and testing of AI functions in mobile apps based on user reviews. Their empirical study on plant identification apps emphasizes the importance of considering user feedback in testing AI functionalities to enhance usability. Furthermore, studies by authors such as Chen et al. (2023), Fan et al. (2022), Brdnik, Heričko & Šumak (2022), Virvou (2023), Kuang et al. (2024) and Hammad et al. (2022) focused on usability evaluation in different contexts, including clinical decision support tools, human-AI collaboration, intelligent user interfaces, the interrelationship between AI and UX, collaboration with conversational AI assistants, and UX design in specialized domains.

Several studies have examined public attitudes and sentiments toward ChatGPT using X (formerly Twitter) data, as shown in Sudheesh et al. (2023), Korkmaz, Aktürk & Talan (2023), Lian et al. (2024) and Nasayreh et al. (2024). These studies encompass sentiment analysis across various contexts, highlighting its performance using innovative models like BERT and identifying research gaps in Arabic sentiment analysis. These studies delve into user opinions, public attitudes, and the accuracy of sentiment classification models, emphasizing the need for further exploration and specialized investigations in this evolving field. Table 1 represents the methodologies used and the results of various recent studies on the usability and user experiences of generative AI tools and their implications for user interaction and satisfaction.

Table 1 Summary of usability studies on generative AI tools.

Reference	Year	Method used	Results	
Pardos & Bhandari (2023)	2023	User interface evaluation and iterative
design human raters	Enhanced usability of ChatGPT due to an intuitive interface and involvement of human raters	
Ren et al. (2022), Ren et al. (2024)	2022	Experimentation and usability testing	Identified ways to optimize chatbots for effective real-time collaboration and user interaction	
Bubaš, Čižmešija & Kovačić (2024)	2024	Introduced assessment scales for usability
and user experiences of conversational AI
tools among university students, focusing
on Bing Chat tool evaluation	Found that usefulness, trust, and design appeal significantly influence students’ intention to use CAI tools	
Mulia, Piri & Tho (2023)	2023	Utilized System Usability Scale (SUS) questionnaire to evaluate quality and usability of ChatGPT	ChatGPT received a SUS score of 67.44, indicating a generally positive user response to its effectiveness, efficiency, and ease of use but highlighted the limitation of data currency	
Skjuve, Følstad & Brandtzaeg (2023)	2023	Surveyed 194 ChatGPT users to explore user experience using questionnaire and thematic analysis within a pragmatic-hedonic framework	User satisfaction with ChatGPT derived from practicality, relevant information, and facilitation of academic or professional tasks, as well as enjoyment and innovative engagement	
Lin (2023)	2023	Design and user interface analysis	Conversational AI can significantly enhance system usability through intuitive user interfaces	
Rodriguez et al. (2024)	2024	Expert evaluation of ChatGPT-generated content	ChatGPT can speed up the development of quality medical technology products and support interdisciplinary communication	
Bandi, Adapa & Kuchi (2023)	2023	Taxonomy and classification of generative AI models, and evaluation metric discussion	Provided insights for implementing and assessing generative AI models effectively	
Romeiro & Araújo (2024)	2024	Development of guideline-based metrics	Contributed to enhancing the usability assessment of mobile interfaces	
Sally (2023)	2023	Conducted text mining and sentiment analysis on Sri Lankan mobile banking apps to understand consumer dissatisfaction	Highlighted the importance of sentiment analysis in evaluating user opinions and improving services based on feedback	
Alshamari & Alsalem (2023)	2023	Conducted a critical review of current issues and trends in usable AI	Emphasized the need for standardized usability measures and further research in AI usability	
Tao et al. (2023)	2024	Explored maintenance and testing of AI functions in mobile apps based on user reviews	Highlighted the importance of considering user feedback in testing AI functionalities to enhance usability	
Chen et al. (2023), Fan et al. (2022), Brdnik, Heričko & Šumak (2022), Virvou (2023), Kuang et al. (2024), Hammad et al. (2022)	2022–2024	Focused on usability evaluation in different contexts, including clinical decision support tools, human-AI collaboration, intelligent user interfaces, interrelationship between AI and UX, collaboration with conversational AI assistants, and UX design in specialized domains	Contributed to advancing the usability of generative AI tools in various domains	
Sudheesh et al. (2023), Korkmaz, Aktürk & Talan (2023), Lian et al. (2024), Nasayreh et al. (2024)	2023–2024	Sentiment analysis across various contexts using X (formerly Twitter) data.	These studies encompass sentiment analysis across various contexts, highlighting its performance using innovative models like BERT and identifying research gaps in Arabic sentiment analysis. These studies delve into user opinions, public attitudes, and the accuracy of sentiment classification models, emphasizing the need for further exploration and specialized investigations in this evolving field.	

While these studies contribute to advancing the usability of generative AI tools, there is a notable research gap. Existing related work primarily relies on expert evaluations, studies, and experiments, neglecting an evaluation of the usability of such tools based on user reviews from digital marketplaces such as App stores or Google play which is a valuable source of data reflecting real-world user experiences. The aim of the current study is to bridge this gap by analyzing user reviews for a nuanced understanding of the usability of generative AI tools. Analyzing such reviews would offer a more comprehensive understanding of the usability of generative AI tools in practical scenarios and provide actionable recommendations for improvement.

Materials and Methods

The materials and methods employed in this study are detailed in this section.

Overview

This study collected 11,549 reviews across Android and iOS platforms. The dataset analyzed in this study comprises reviews from users of generative AI apps offered by five applications, namely ChatGPT (https://openai.com/gpt-4), Bing AI (https://www.bing.com/search), Microsoft Copilot (https://www.microsoft.com/en/copilot), Gemini AI (https://gemini.google.com), and Da Vinci AI (https://davinci.ai). These reviews are publicly available in a repository. The research methodology employed sentiment analysis to evaluate English reviews. Unique to this method is its ability to process large datasets without needing to identify participants or develop specific analytical tools like questionnaires or personal interviews, offering a direct and unrestricted platform for user expression. In contrast, methods such as SUS and UEQ require the development of specific evaluation tools. An overview of the research approach is depicted in Fig. 1.

Figure 1 Review analysis schema.

This figure illustrates the schema used for analyzing user reviews of generative AI applications. It details the steps involved in data collection, preprocessing, sentiment analysis, and the extraction of key insights from the user feedback.

Data collection and processing

Data collection: The dataset collected in this study comprises reviews from users of generative AI apps offered by five applications, namely ChatGPT, Bing AI, Microsoft Copilot, Gemini AI, and Da Vinci AI. A combination of Python (https://www.python.org) scripts and web scraping tools, including Google Play Scraper (https://www.npmjs.com/package/google-play-scraper), App Store Scraper (https://www.npmjs.com/package/app-store-scraper), and Octoparse (https://www.octoparse.com), were employed to collect 11,549 reviews across Android and iOS platforms over a period from January to March 2024. The Android users contributed 4,776 reviews, while iOS users provided 6,773.

Data cleaning: The collected data underwent a cleaning process utilizing the Pandas (https://pandas.pydata.org) and numPy (https://numpy.org) libraries in a Jupyter (https://jupyter.org) Notebook environment. In this study, the following preprocessing steps were applied to the user reviews:

• HTML tag removal: HTML tags were removed from the user reviews using regular expressions.

• Stop word elimination: Stop words, such as “the”, “a”, and “is”, were removed using the NLTK library’s nltk.corpus.stopwords function.

• Tokenization: The text was split into individual words or phrases called tokens using the NLTK library’s nltk.tokenize.word_tokenize function.

• Normalization: Tokens were converted to a common form, such as lowercase or lemmatization, using the NLTK library’s nltk.stem.WordNetLemmatizer function.

• Punctuation and special character removal: Punctuation marks, special characters, and duplicates were removed from the dataset.

After that, the data was consolidated into a single DataFrame and subsequently converted into a CSV file. The cleaning phase resulted in 3,958 Android reviews and 3,778 iOS reviews, totaling 7,736 reviews, ensuring the data’s suitability for analysis.

Figure 2 summarizes the total reviews collected from all the apps across the Apple’s App Store and Google Play, indicating that the ChatGPT app received the highest number of reviews in the Apple Store. It is noted that there are no reviewers for the Gemini app in the Apple’s App Store, attributed to the absence of the Gemini version in the store.

Figure 2 Total reviews collected from generative AI apps on Apple’s App Store and Google Play.

This bar chart shows the dataset collected in this study, comprising reviews from users of five Generative AI applications: ChatGPT, Bing AI, Microsoft Copilot, Gemini AI, and Da Vinci AI. The dataset includes over 11,000 reviews gathered from both Android and iOS platforms, spanning the period from January to March 2024. It provides an overview of the volume of user feedback received for each app.

This dataset includes various features such app names, descriptions, ratings, and review desciption and review date. The dataset is publicly available. Figure 3 provides a snapshot of the data.

Figure 3 Snapshot of our dataset for generative AI user reviews across Android and iOS platforms accessible on Kaggle.com.

This figure provides a snapshot of the dataset used for the analysis of user reviews. The dataset is available on Kaggle.com and includes user reviews for generative AI apps on both Android and iOS platforms. The dataset includes information on the app name, platform, review text, and review date.

Lexicon construction and sentiment analysis

Polarity Lexicons: In order to differentiate among usability factors, namely satisfaction, effectiveness, and efficiency, we constructed three polarity lexicons using the NLTK library (https://www.nltk.org). These lexicons were populated with opinion words derived from the reviews and contained a mix of positive and negative words corresponding to each factor. The satisfaction lexicon consisted of 110 words, with 60 positives and 50 negatives. The effectiveness lexicon comprised 76 words, including 47 positives and 29 negatives. Lastly, the efficiency lexicon contained 23 words, with 10 positives and 13 negatives. Table 2 provides a summary of the polarity distribution among the usability factors, demonstrating that the satisfaction factor contained more positive words, while the efficiency factor had a higher proportion of negative words. Furthermore, a sample of positive and negative polarity words is presented in Table 3.

Table 2 Summary of polarity distribution among the usability factors.

Usability factors	No. of positive polarities	No. of negative polarities	
Satisfaction	60 (54.5%)	50 (46.5%)	
Effectiveness	47 (61.84%)	29 (38.15%)	
Efficiency	10 (43.47%)	13 (56.52%)	

Table 3 Sample of positive and negative polarity words.

Satisfaction Lexicon	Effectiveness Lexicon	Efficiency Lexicon	
Positive	Negative	Positive	Negative	Positive	Negative	
Good	Bad	Effective	Error	Quick	Wrong	
Excellent	Not happy	Working perfectly	Slow	Fast	Lagging	
Nice	Unsatisfied	Reliable	Unreliable	Correct	Incorrect	

Sentiment analysis: The NLTK library in Python was employed to label reviews of each generative AI app. The model assessed usability scores based on three factors: effectiveness, efficiency, and satisfaction. The Vader model (Hutto & Gilbert, 2014), a sentiment analysis tool, from NLTK provided polarity scores reflecting the sentiment expressed in each review, with the scoring scale ranging from −1 (negative sentiment) to +1 (positive sentiment). Negative values indicated negative opinions, zero represented neutral opinions, and positive values indicated positive opinions. The Vader model, implemented through the nltk.sentiment.vader package in Python, evaluates the sentiment of each text by calculating a compound score. The compound score is a normalized, weighted composite score that ranges from −1 to +1, representing the overall sentiment of the text. The process involved the following steps:

• Importing the VADER sentiment analyzer: from nltk.sentiment.vader import SentimentIntensityAnalyzer

• Initializing the sentiment intensity analyzer: sia = SentimentIntensityAnalyzer()

• Applying the model to the text data: sentiment_scores = sia.polarity_scores(text) The sentiment scores include four components: ‘neg’ (negative), ‘neu’ (neutral), ‘pos’ (positive), and ‘compound’ (overall sentiment). For this analysis, the compound score was primarily used to determine the sentiment polarity and intensity of each review.

The VADER model was used with its default parameters, which were found to be effective for our dataset. These parameters include:

• lexicon: The VADER sentiment lexicon.

• norm_scalar: A scaling factor used to normalize the compound score.

• compound_rule_mod: A modifier used to adjust the compound score based on the presence of certain grammatical structures.

Usability/compound scoring

Each review’s total usability/compound score was determined by averaging the individual scores of satisfaction, effectiveness, and efficiency. The overall mean of these scores was then computed, yielding a score range from -3 (least usable) to +3 (most usable). The methodology for assigning polarity scores to words found in the reviews is demonstrated in Table 4, which aligns with the constructed lexicons.

Table 4 Polarity scoring of lexicon words based on usability factors.

Review	Satisfaction	Effectiveness	Efficiency	Usability/compound score	
1. In Bing News, there is no way to translate it, in Google News, there is an option to translate it just like I do on the browser. Please add basic features it’s a shame.	−1	−1	−1	−3	
2. I don’t like that I’m forced to replace Google Assistant with Gemini to use this app at all. I just wanted the app to have another GPT to try out. It’s not even a functional assistant. It doesn’t let me play music on Spotify.	−1	−1	−1	−3	
3. Love the rewards incentive. Really appreciate the opportunities to earn reward points. Enjoy the free variety of photographs for my phone’s lock and menu screens. Satisfying experience with the search engines’ results. Maintenance updates keep everything flowing smooth and secure	+1	+1	+1	+3	
4. Very very bad experience, if anyone wants download this app please I request to you don’t waste your time and data with this app.	−1	−1	−1	−3	
5. works fantastic it’s just not perfect yet still have some minor problems with it.	+1	+1	−1	+1	

Results and Analysis

Results

After arranging the reviews into separate Excel sheets for each operating system, we categorized and determined the degree of each compound factor. The statistics presented in Fig. 4 regarding users’ reviews of generative AI apps from Apple’s App Store indicate that the ChatGPT app received the highest percentage of positive reviews (57%) and the highest percentage of negative reviews (52%). This dual distinction can be attributed to the fact that ChatGPT garnered a larger number of reviews compared to other apps, as previously illustrated in Fig. 2.

Figure 4 Comparative analysis of user reviews for generative AI apps on Apple’s App Store.

This figure presents a comparative analysis of user reviews for generative AI apps available on the Apple App Store. It shows the percentage distribution of positive and negative reviews for each app, highlighting user sentiment towards different generative AI applications. Note that Gemini reviews are not included in this analysis due to the absence of the Gemini version in the Apple App Store.

The data presented in Fig. 5 illustrates that the ChatGPT app received the highest percentage of positive reviews (34%), while Gemini had the highest percentage of negative reviews (40%) among users of generative AI apps on Google Play. This discrepancy highlights the need for further investigation into the reasons behind Gemini’s negative feedback, especially considering that only 12% of reviewers originated from this app, as shown in Fig. 2.

Figure 5 Comparative analysis of user reviews for generative AI apps on Google Play.

This figure presents a comparative analysis of user reviews for generative AI apps available on Google Play. It shows the percentage distribution of positive and negative reviews for each app, highlighting user sentiment towards different generative AI applications.

Tables 5 and 6 show the following data for Android and iOS, respectively: the name of the app, the number of reviews, the polarity weight of positive and negative reviews, neutral reviews, and the overall compound score.

Table 5 Compound score of android users.

App	Total
reviews	Positive
reviews
(PR)	Negative
reviews
(NR)	Neutral
reviews
(NER)	Satisfaction
Score	Effectiveness
Score	Efficiency
score	Compound/
usability
score	
ChatGPT	1,393	1,254	71	68	0.590	0.479	0.443	0.504	
Bing AI	795	585	152	58	0.427	0.416	0.354	0.399	
Microsoft Copilot	1,191	1,019	108	64	0.530	0.444	0.424	0.466	
Da Vinci	795	585	152	58	0.402	0.410	0.364	0.392	
Gemini	595	231	316	48	−0.138	0.108	0.078	0.016	

Table 6 Compound score of iOS users.

App	Total
reviews	Positive
reviews
(PR)	Negative
reviews
(NR)	Neutral
reviews
(NER)	Satisfaction
score	Effectiveness
score	Efficiency
score	Compound/
usability
score	
ChatGPT	2,102	1,578	414	110	0.565	0.343	0.478	0.462	
Bing AI	320	223	76	21	0.373	0.317	0.301	0.330	
Microsoft Copilot	690	512	120	58	0.438	0.313	0.454	0.401	
Da Vinci	680	445	188	47	0.319	0.258	0.248	0.275	

In this analysis, the compound scores were derived by averaging the satisfaction, effectiveness, and efficiency scores. Table 5 presents the compound scores of all generative apps on Android, while Table 6 displays the compound scores for all generative AI Apps on the iOS system. For Android users, Notably, ChatGPT secured the highest positive compound score at 0.504, with Microsoft Copilot trailing closely at 0.466. Conversely, Gemini AI registered the lowest, with a compound score of 0.016. Moving to iOS users, here, ChatGPT also leads with a compound score of 0.462, followed by Microsoft Copilot at 0.401. The Da Vinci app recorded the lowest score in this category, at 0.275.

The analyses presented in Tables 5 and 6 underscore the pivotal role of satisfaction scores in gauging usability, supported by effectiveness and efficiency factors. Among Android users, the ChatGPT app demonstrated the highest satisfaction rate at 0.590, followed by the Copilot app with a score of 0.530. Conversely, the Gemini app had the lowest Satisfaction rate among all applications, with a calculated rate of −0.138. Among iOS users, the ChatGPT app had a Satisfaction rate of 0.565, followed by the Microsoft Copilot app with a score of 0.438. Additionally, the Da Vinci app had the lowest satisfaction rate, calculated at 0.275. Furthermore, Figure 6 displays the compound scores of satisfaction, effectiveness, and efficiency for generative AI apps derived from reviews collected from both Apple’s app stores and Google Play. These scores represent an aggregated measure of user feedback across three dimensions: satisfaction, effectiveness, and efficiency, providing a holistic view of the overall usability of each app across both platforms. It is evident from the figure that ChatGPT boasts the highest compound score of 0.483, followed by Microsoft Copilot with the second-highest score of 0.434. Note that Gemini reviews are not included in this figure due to the absence of the Gemini version in the Apple App Store.

Figure 6 Comparative compound scores of generative AI apps based on user satisfaction, efficiency, and effectiveness across both stores.

The figure displays the compound scores of satisfaction, effectiveness, and efficiency for generative AI apps collected from both Apple’s App Store and Google Play. These scores represent an aggregated measure of user feedback across three dimensions: satisfaction, effectiveness, and efficiency, providing a holistic view of the overall usability of each app across both platforms. Note that Gemini reviews are not included in this figure due to the absence of the Gemini version in the Apple App Store.

Our analysis generally matches the digital store average reviews, reflecting similar trends in user satisfaction and usability scores. Table 7 presents the average ratings from the App Store and Google Play alongside the compound usability scores for both iOS and Android users. The compound usability scores are derived from metrics of satisfaction, effectiveness, and efficiency, highlighting the overall user experience and performance of each app across different platforms. As shown in the table, there is a general alignment between the average review ratings and the compound usability scores. ChatGPT, which received the highest compound usability score, also has the highest average ratings on both the App Store and Google Play. However, there are some notable discrepancies. For example, Gemini AI has a higher average rating on the Google Play (4.6) than Da Vinci AI (4.2), but a lower compound usability score. This suggests that users may be more likely to leave positive reviews for Gemini AI on the Google Play, even if they have mixed feelings about the app’s usability. The absence of data for iOS further highlights its limited user base or lack of availability on that platform. Overall, ChatGPT and Microsoft Copilot stand out as the top-performing generative AI apps in terms of both user ratings and compound usability scores, indicating a high level of user satisfaction and effectiveness across both platforms. In contrast, Gemini AI, despite having a decent rating on Google Play, shows a significant usability gap on Android, suggesting areas for improvement.

Table 7 Comparison of average ratings and compound usability scores for generative AI apps on iOS and Android platforms.

App name	Average rating
(App Store)	Compound
usability
score
(iOS users)	Average rating
(Google Play)	Compound
usability
score
(Android users)	
ChatGPT	4.8	0.462	4.7	0.504	
Microsoft Copilot	4.8	0.401	4.6	0.466	
Bing AI	4.7	0.330	4.6	0.399	
Da Vinci AI	4.6	0.275	4.2	0.392	
Gemini AI	NA	NA	4.6	0.016	

Discussion

This study presents the first systematic evaluation of the usability of generative AI applications using ISO 9241 usability standards. By analyzing 11,549 user reviews from five generative AI apps, our findings reveal several key insights into the usability challenges and opportunities associated with these emerging technologies. Ultimately, the study provides specific recommendations for improving the usability of generative AI applications, thereby contributing to informed decision-making for developers, researchers, and businesses relying on generative AI solutions.

The quantitative analysis of generative AI applications has highlighted several core issues affecting their perceived usability. Misinformation stands out as a prevalent issue, Our analysis revealed that the five generative AI applications frequently provided incorrect information. Negative reviews highlighted instances where the AI-generated answers were found to be inaccurate and erroneous. In terms of response times, the analysis indicates a clear disparity between user expectations and app performance. Applications such as ChatGPT and Bing often have prolonged response times. For instance, rephrasing a 500-word essay could take as long as 10 min. This finding aligns with the results of the study mentioned in Lee et al. (2024), which indicates that ChatGPT-3.5 and Bing Chat exhibit extended response times compared to ChatGPT-4. Users also reported server-related delays, exemplified by a 16-minute wait for Da Vinci AI to generate a an image of a cat. The relevance of content delivered by generative AI apps is also called into question. It is essential for generative AI apps to deliver pertinent information. User reviews, however, frequently cite the provision of incorrect information, especially in historical and scientific contexts, which can misguide users. Image generation, a touted feature of these AI tools, also falls short of user expectations. Both Da Vinci and Gemini AI have struggled to produce relevant and requested images, leaving users dissatisfied with the output. For example, Gemini AI faced challenges in generating accurate depictions of specific human figures and historical images.

Finally, the feature set and functionality of apps like ChatGPT 3.5 and Gemini are noted to be insufficient, with users highlighting Gemini’s subpar performance compared to more established tools like Google Assistant. This finding is consistent with the results reported in Rane, Choudhary & Rane (2024), which indicates that users have observed Gemini’s inferior performance compared to established tools like Google Assistant. Frustrations also arise from the limitations present within the free versions of these apps, suggesting a need for more generous feature access to improve user engagement and satisfaction.Based on the analysis of positive and negative user reviews of generative AI apps, several recommendations are provided to improve usability. Firstly, it is recommended to enrich the AI’s mathematical problem-solving capabilities by training it with a diverse set of math problems to improve accuracy. Secondly, developers should ensure that AI models are trained with comprehensive and verified data. This measure aims to minimize instances of misinformation and inaccuracies in the generated responses. Improving network latency and investing in robust servers is another crucial recommendation. This can significantly reduce response times of generative AI apps, ensuring quicker and more seamless interactions with users.The analysis also highlights the importance of utilizing user reviews as valuable feedback for continuous improvement. By actively considering user feedback and incorporating it into the development process, developers can enhance the overall user experience. Furthermore, enhancing customer support is essential to boost user satisfaction. It is crucial for developers to actively respond to user queries and concerns, aiming for a higher response rate. During the review collection process, it was observed that only a small percentage of user queries received responses. Lastly, the introduction of new features, including within the free versions of the apps, may expand the user base and potentially elevate app ratings. Implementing these improvements should be done in accordance with user needs and expectations, which is crucial for increasing the overall effectiveness and adoption of generative AI applications.

Limitations and future work

While the research has provided valuable insights, several limitations have been identified that warrant consideration for future work. Firstly, the dynamic nature of the app market may render the study findings obsolete as apps undergo continuous updates and improvements. This limitation underscores the need for ongoing assessments to capture the evolving landscape of generative AI apps. Future research should consider conducting longitudinal studies to track changes in usability over time and keep pace with app updates (Tao & Edmunds, 2018). Secondly, the absence of considerations regarding personalization and cultural relevance within user experiences represents a limitation. Future studies could explore how these factors influence user interactions with generative AI apps, providing a more comprehensive understanding of user preferences and behaviors across different demographics and cultural backgrounds (Shen et al., 2020). Additionally, expanding the analysis beyond English reviews to include a wider range of languages can offer a more inclusive representation of the global user base and enhance the applicability of the findings on a broader scale. One limitation of this study is the lack of distinction between reviews from subscribed and unsubscribed users. Subscribed users have access to additional features and capabilities, such as longer response lengths, faster response times, and access to exclusive models. These differences could potentially influence user sentiment and feedback. To address this limitation in future research, we recommend collecting data that specifically categorizes reviews based on user subscription status. This could involve direct data collection from platforms that offer subscription details or integrating user surveys to differentiate between these groups. Additionally, we will explore the possibility of using machine learning techniques to identify patterns in user reviews that may be indicative of subscription status.

Another limitation is the potential bias in online reviews. Users who provide feedback may have extreme experiences, either very positive or negative, which could skew the overall sentiment analysis results. Future research could explore strategies to mitigate review biases, such as incorporating a larger and more diverse sample of user reviews to ensure the sample includes reviews from users with different demographics, and backgrounds. Additionally, using random sampling techniques to select a representative sample of reviews can help achieve this goal. Moreover, employing advanced sentiment analysis techniques that account for varying degrees of sentiment intensity and identify nuanced emotions beyond simply positive or negative sentiments can further mitigate bias (Ferrara, 2024). For example, models like Bidirectional Encoder Representations from Transformers (BERT) can help detect extreme sentiments more accurately and classify them accordingly. Furthermore, developing algorithms to identify and filter out reviews that are clearly promotional, malicious, or irrelevant is another potential strategy. Review normalization, where different weights are assigned to reviews based on their content quality and the credibility of the reviewers, can help mitigate the influence of biased or extreme reviews. For instance, reviews from verified purchasers can be given more weight compared to anonymous reviews.

Lastly, the Valence Aware Dictionary and sEntiment Reasoner (VADER) model, while effective for basic sentiment analysis, has several limitations. As a rule-based sentiment classifier, VADER relies on predefined lexicons and rules, which limits its ability to capture nuanced sentiments and context-specific meanings accurately. This approach can fail to interpret sarcasm and irony, leading to potential misclassifications. Additionally, VADER’s effectiveness can be compromised by misspellings and grammatical errors in the text, which may cause it to overlook significant words or misinterpret the sentiment. Its reliance on English language text further restricts its applicability in multilingual contexts. Despite its advantages, such as simplicity and computational efficiency, VADER’s inability to adapt to complex linguistic variations and its dependency on predefined sentiment rules highlight its limitations in providing a comprehensive sentiment analysis (Kim, Spörer & Handschuh, 2024). To overcome VADER’s limitations, future research can focus on incorporating contextual information using NLP techniques. Also, expanding language support which could involve training models on datasets of text in different languages or using techniques such as machine translation to translate text into a language that the model can understand. Moreover, improving explainability by integrating LIME (Local Interpretable Model-Agnostic Explanations) to explain the model’s predictions in a human-understandable way could enhance VADER’s effectiveness.

In light of the identified limitations, the findings of the study suggest key areas for improvement in information delivery, response times, feature set, and customer support. Future studies should aim to incorporate a wider range of languages, employ more sophisticated sentiment analysis tools, and implement strategies to mitigate biases in online reviews, thereby advancing the understanding of user sentiment and usability evaluation of generative AI apps.

Conclusions

The analysis of user sentiment and usability evaluation of generative AI apps, focusing on ChatGPT, Bing AI, Microsoft Copilot, Gemini AI, and Da Vinci AI, has shed light on critical aspects impacting user experiences. The results show that ChatGPT leads in usability scores for both Android and iOS in user reviews of generative AI apps. Android’s lowest score is for Gemini, while iOS’s lowest is Da Vinci. Satisfaction significantly influences the overall usability score. The analysis of user reviews revealed challenges such as misinformation, extended response times, relevance of content, image generation issues, and insufficient feature sets in certain apps.

The impact and novelty of our findings lie in highlighting specific usability challenges and user concerns that are often overlooked in controlled experimental studies. By systematically analyzing user feedback across multiple generative AI applications, we offer a unique perspective on the practical implications and areas for improvement in these tools. This approach underscores the importance of user-centered design and continuous refinement based on user feedback, which is crucial for the evolution of generative AI technologies.

This study not only highlights specific usability challenges but also offers practical recommendations that can directly inform future development efforts in the generative AI domain. By addressing these challenges, developers can significantly improve user satisfaction and the overall effectiveness and adoption of generative AI applications. Future research can build on our findings by empirically testing the proposed improvements and exploring additional factors that influence the usability and acceptance of generative AI tools, thereby contributing to the ongoing evolution of these technologies.

Additional Information and Declarations

Competing Interests

Author Contributions

Data Availability

The author declares that they have no competing interests.

Reham Alabduljabbar conceived and designed the experiments, performed the experiments, analyzed the data, performed the computation work, prepared figures and/or tables, authored or reviewed drafts of the article, and approved the final draft.

The following information was supplied regarding data availability:

The data are available at Kaggle and Zenodo:

- https://www.kaggle.com/datasets/rehamalabduljabbar/gen-ai-tools-appstore-googleplay/data.

- Alabduljabbar, R. (2024). Gen AI apps user review from AppStore GooglePlay [Data set]. Zenodo. https://doi.org/10.5281/zenodo.12665333.

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
