# Peer review of "User-centric AI: evaluating the usability of generative AI applications through user reviews on app stores"

_PeerJ Computer Science, doi:10.7717/peerj-cs.2421_

## Round 0.1 · original submission · Major Revisions

The authors must improve the paper accordingly to the reviewers comments.

Reviewer 1 ·

Basic reporting

The paper presents a usability evaluation of generative AI applications by analyzing user reviews from Apple’s App Store and Google Play. The study covers five AI apps: ChatGPT, Bing AI, Microsoft Co-Pilot, Gemini AI, and Da Vinci AI, using ISO 9241 usability standards. The findings indicate that ChatGPT had the highest usability scores, while Gemini AI scored the lowest. The study concludes with recommendations to improve the usability of these tools.
1. The paper is generally well-written, but there are some typos. For example, line 96 should be “Background”. Please carefully double-check the spelling and grammar.
2. The author provided a comprehensive introduction and literature review. However, the discussion about interpreting the results from this paper did not highlight the novelty of the current study. It is recommended that the author start the discussion section with a paragraph to summarize and highlight the novelty of the current study.
3. The author should suggest/explore methods to mitigate the biases and limitations mentioned in the paper's discussion section.
4. No code is submitted for data collection.
5. Since the current study solely relied on user reviews, those might not provide a comprehensive picture of usability issues. The author should suggest a future direction to address this issue.
6. Are the reviews for ChatGPT distinguishable between subscribed and unsubscribed users? Could that be a limitation of the study?
7. Do the results from the current analysis match the digital store average review?

Experimental design

The research question is well-defined and addresses a significant gap in the literature on the usability of generative AI tools based on real-world user reviews. The methodology is clearly described, with sufficient detail to replicate the study. Using sentiment analysis to evaluate user reviews is appropriate for the research question. However, there are some limitations, as also mentioned above. The study relies solely on user reviews from digital marketplaces. While this provides valuable insights, it may introduce bias as these reviews can be influenced by factors unrelated to usability, such as marketing efforts or brand loyalty. Additionally, the use of the VADER model for sentiment analysis, though appropriate, is not discussed in depth regarding its limitations. For instance, VADER may not capture nuanced sentiments or context-specific meanings accurately.

Validity of the findings

The large and publicly available dataset enhances the study's transparency and reproducibility. The statistical methods used for analyzing the data are sound, and the study provides a clear explanation of how usability scores were calculated. The conclusions are well-stated and linked to the original research question. However, the study's actionable recommendations could benefit from further validation through empirical testing.

Additional comments

1. In Figure 4, I don’t see the reviews from Gemini included in the analysis. If it’s excluded, please state the reason in the figure as well.
2. In Figure 6, I don’t think a stack bar graph is the correct visualization in this context because the numbers do not add up to 1. A bar graph to separate each GPT-like app would be better.
3. The author should provide more descriptive figure legends.
4. For the tables, I am confused as to why all the tables are labeled as “Table 1”.

Reviewer 2 ·

Basic reporting

1. The writing is clear and professional, but the paper's focus on market research/analysis with NLP techniques may not fully align with the journal's scope of computer science.
2. The literature references and field background/context are sufficient, but more details about the sentiment analysis techniques used would be beneficial.
3. The article structure is professional, but some figures (e.g., Figure 4) could be improved to avoid visual misleading.
4. The paper analyzes user reviews, not directly testing a hypothesis. The results are self-contained.

Experimental design

1. The research question is well-defined and relevant, but the investigation could be more rigorous in terms of explaining the sentiment analysis techniques used and providing more detailed results.
2. The research question of evaluating user sentiment towards Generative AI apps is well-defined, but might fall more under market research than core computer science.
3. The methodology using NLTK for sentiment analysis is mentioned but lacks details for replication.

Validity of the findings

1. The conclusions are well-stated, but the impact and novelty of the findings are limited.
2. The paper links review sentiment to reasons like response time. Conclusions could be more convincing if the paper provided more robust data analysis and supporting data (e.g., response time comparison, user attitude).

Additional comments

1. The paper focuses on user sentiment analysis of Generative AI apps, which leans more towards market research with NLP techniques. While the use of NLTK is relevant to computer science, the core contribution might be a better fit for a marketing research or user experience research journal.
2. Figure 4 could be redundant as they seem to repeat information from Figure 2. Standardizing pie chart sizes would reduce misleading information.
3. Overall, the paper could benefit from a more in-depth analysis of the sentiment data and stronger connections between user reviews, sentiment scores, and user experience factors.

---

## Round 0.2 · Minor Revisions

Some final minor revisions must be done.

Reviewer 1 ·

Basic reporting

The author addressed my concern carefully, and I recommend that the journal accept the paper for publication. A few things I want to bring to the author’s attention:
1. I don’t think keeping the username in the dataset would add additional information to the conclusion. I recommend removing the username from your dataset to prevent potential privacy policy violations. There are unique identifiers for each review already.
2. There are some formatting issues in the revised manuscript.

Experimental design

The author addressed my concern carefully, and I recommend that the journal accept the paper for publication.

Validity of the findings

The author addressed my concern carefully, and I recommend that the journal accept the paper for publication.

Additional comments

The author addressed my concern carefully, and I recommend that the journal accept the paper for publication.

Reviewer 2 ·

Basic reporting

The author has successfully adjusted the language and view of the paper, making it more suitable for a computer science journal, specifically in the area of user experience and HCI. The literature references, field background, and context are sufficient and well-provided. The article structure, figures, tables, and raw data sharing remain professional and well-organized. Overall, the paper is self-contained, presenting relevant results to the hypothesis, with formal results including clear definitions of all terms and theorems, and detailed proofs.

Experimental design

The author has successfully aligned the paper with the Aims and Scope of the journal, presenting original primary research in the area of user experience and HCI. The research question is well-defined, relevant, and meaningful, filling an identified knowledge gap in the analysis of user reviews of Generative AI apps. The investigation is rigorous, performed to a high technical and ethical standard. The methods, including sentiment analysis techniques, are described with sufficient detail and information to replicate.

Validity of the findings

The author has provided robust and statistically sound data, although some minor issues remain, such as the scores of three dimensions being all -1 in Table 4. This minor issue notwithstanding, the conclusions are well-stated, linked to the original research question, and limited to supporting results. The author has addressed my previous concerns regarding the lack of analysis depth, providing more details on sentiment analysis techniques.

Additional comments

I am pleased to report that the author has successfully addressed my previous concerns, and the paper has greatly improved since the last version. One minor suggestion I have is that the author provides more diverse samples in Table 4, with varying scores on the three dimensions, to better illustrate the data points. Overall, the author has made significant improvements.

---

## Round 0.3 · accepted · Accept

The paper can be accepted. It was very well improved.

Reviewer 2 ·

Basic reporting

The paper is well-written in clear and unambiguous, professional English. Literature references and field background/context are now sufficient. The article structure, figures, tables, and raw data sharing are professional.

Experimental design

The research question is well-defined and relevant, and the paper clearly states how it fills an identified knowledge gap. The investigation is rigorous and performed to a high technical and ethical standard. The methods are described in sufficient detail to replicate.

Validity of the findings

The data provided is robust, statistically sound, and controlled. The conclusions are well-stated and linked to the original research question.

Additional comments

The author has addressed all my previous concerns.